# A Human Stem Cell-Derived Neurosensory–Epithelial Circuitry on a Chip to Model Herpes Simplex Virus Reactivation

**DOI:** 10.3390/biomedicines10092068

**Published:** 2022-08-24

**Authors:** Pietro Giuseppe Mazzara, Elena Criscuolo, Marco Rasponi, Luca Massimino, Sharon Muggeo, Cecilia Palma, Matteo Castelli, Massimo Clementi, Roberto Burioni, Nicasio Mancini, Vania Broccoli, Nicola Clementi

**Affiliations:** 1Division of Neuroscience, San Raffaele Scientific Institute, 20132 Milan, Italy; 2Laboratory of Microbiology and Virology, Vita-Salute San Raffaele University, 20132 Milan, Italy; 3Department of Electronics, Information and Bioengineering, Politecnico di Milano, 20133 Milan, Italy; 4Laboratory of Microbiology and Virology, IRCCS San Raffaele Hospital, 20132 Milan, Italy; 5National Research Council (CNR), Institute of Neuroscience, 20129 Milan, Italy

**Keywords:** reactivation, latency, herpes simplex virus, organoids, microfluidics, keratinocytes

## Abstract

Both emerging viruses and well-known viral pathogens endowed with neurotropism can either directly impair neuronal functions or induce physio-pathological changes by diffusing from the periphery through neurosensory–epithelial connections. However, developing a reliable and reproducible in vitro system modeling the connectivity between the different human sensory neurons and peripheral tissues is still a challenge and precludes the deepest comprehension of viral latency and reactivation at the cellular and molecular levels. This study shows a stable topographic neurosensory–epithelial connection on a chip using human stem cell-derived dorsal root ganglia (DRG) organoids. Bulk and single-cell transcriptomics showed that different combinations of key receptors for herpes simplex virus 1 (HSV-1) are expressed by each sensory neuronal cell type. This neuronal–epithelial circuitry enabled a detailed analysis of HSV infectivity, faithfully modeling its dynamics and cell type specificity. The reconstitution of an organized connectivity between human sensory neurons and keratinocytes into microfluidic chips provides a powerful in vitro platform for modeling viral latency and reactivation of human viral pathogens.

## 1. Introduction

Herpes simplex viruses (HSVs) have the characteristic of establishing long-term latency in peripheral neurons, from which they reactivate, causing symptoms in innervated tissues. The frequency and severity of reactivations vary according to the characteristics of the host’s immune response and of the virus isolate involved. Even if some essential molecular mechanisms have already been clarified [1,2], reactivation studies of neurotropic viruses still represent a challenge regarding obtaining a fully human, highly reproducible model with the specific cells involved in virus tropism. In vivo experiments have been described, but they are endowed with the limitations of using a human-adapted pathogen in animal model systems [3,4,5,6].

The first aspect to be addressed to solve this limitation is a convenient human neuronal system for a reliable HSV reactivation protocol. Different approaches have been described so far. HSV infection has been tested in several experimental settings [7,8,9,10], using differentiated and undifferentiated human neuroblastoma-derived cell lines and subsequent reactivation using thermal or chemical stimuli. Furthermore, complex systems have been developed using human induced pluripotent stem cell (hiPSC)-derived neurons [11]. It has been observed that, while useful for studying the dynamics of viral expression, these cultures do not reproduce all the complexities of HSV-1 reactivation in vivo. Following the advent of 3D cellular cultures, designated organoids, these models have also been investigated [12]. For brain organoids, the 3D culture systems mimic aspects of brain architecture and allow complex cell–cell communication, cell–cell interaction, and cell–extracellular matrix interaction. Brain organoids have given a great impulse to the study of other neurotropic viruses, including the Zika virus and VZV [13,14]. Recent data describe how they can be efficiently infected by HSV-1, but the treatment of latently infected organoids with several stimuli has not led to viral reactivation [12].

The second issue is to reconstitute the neural connection with the cells affected by productive infection to allow axonal transport and neuron-to-cell spread. This issue is crucial for studying HSV infection. Recently, in vitro methods have been developed using pseudorabies virus (PRV) and HSV [12,15,16,17]. The methods center on a microfluidic chamber system that directs the growth of axons into a fluidically isolated environment and recapitulates all known aspects of neuron-to-cell spread. Microfluidic chip devices indeed have been rapidly improved and commercialized (such as Xona Microfluidics devices/chips). Their high versatility and customization allow the reproduction of specific environments that mimic the differences between human tissue compartments, to dissect physiological conditions that could only be analyzed in vivo [18]. Results were achieved on VZV reactivation from latency by using hESC-derived neurons, but no neuron-to-cell connection was investigated [19]. Therefore, induction and reactivation from latency of herpesviruses on human DRGO neurons through infection of primary keratinocytes synaptically connected with them have only been partially explored using human 2D culture [17].

In this study, we took advantage of human iPSC-derived 3D dorsal root ganglion-like organoids (DRGOs), generated by in vitro differentiation of hiPSCs [20], to adapt previous models in a fully human 3D setting. Molecular and genomic analyses and topographic connectivity to peripheral targets have confirmed that DRGOs are populated by nociceptive, mechanoreceptive, and proprioceptive neurons. Herein, we have moved forward by establishing a fully human circuit between DRGO-derived sensory neurons and primary keratinocytes. Furthermore, the circuit was organized in a patterned topography within a miniaturized microfluidics platform specifically optimized for 3D culture [20], thus making it extremely feasible for its reproducibility and translational application. We confirmed the neuronal expression of the cardinal HSV-1 receptors, and, finally, we assessed a functional protocol to study HSV-1 latency and reactivation in vitro. 

## 2. Materials and Methods

**Cell lines**. Vero E6 (Vero C1008, clone E6—CRL-1586; ATCC, Manassas, VA, USA), SH-SY5Y (CRL-2266; ATCC, Manassas, VA, USA), and HaCaT (kindly provided by Cremona O. [21]) cells were cultured in Dulbecco’s Modified Eagle Medium (DMEM) supplemented with non-essential amino acids (NEAAs), penicillin/streptomycin (P/S), HEPES buffer, and 10% (*v*/*v*) fetal bovine serum (FBS). Primary NHEK cells from two adult donors (C-12003 lot numbers #401Z028.1 and #415Z005.2; PromoCell, Heidelberg, Germany) were cultured in Keratinocyte Growth Medium 2 (C-20011; PromoCell, Heidelberg, Germany). Human iPSCs from healthy control fibroblasts (DIGI and Neof2, obtained from the IRCCS Carlo Besta Neurological Institute, Milan, Italy and ATCC, Manassas, VA, USA, respectively) were reprogrammed using the CytoTune-iPS 2.0 Sendai Reprogramming Kit (A16517; Thermo Fisher Scientific, Waltham, MA, USA), maintained in feeder-free conditions in an mTeSR1 (85850; Stem Cell Technologies, Vancouver, BC, Canada) medium supplemented with 1% penicillin/streptomycin (P0781; Merck, Darmstadt, Germany) in 6-well culture plates coated with Matrigel hESC-Qualified Matrix (354277; Corning, Corning, NY, USA).

**Antiviral compound**. Acyclovir (ACV; 9-[(2-hydroxyethoxymethyl) guanine]) (A-4669; Merck, Darmstadt, Germany) was dissolved in DMSO at a concentration of 10 mg/mL and stored as single-use aliquots at −20 °C. Dilutions were made in cell medium immediately before use.

**Generation of 3D DRG-like organoids (DRGOs)****.** iPSC-derived DRGOs were developed as follows: First, 9 × 10^3^ iPSCs were seeded into low-adhesion V-bottom 96-well plates (277143; Thermo Fisher Scientific, Waltham, MA, USA). The following day, the medium was changed to KSR medium (DMEM-F12 with 15% KSR, 1% of each: penicillin/streptomycin, non-essential amino acids, and β-mercaptoethanol 100 μM and L-glutamate 2 nM). The day after (DIV 0), KSR medium plus SB431542 10 μM (S4317; Merck, Darmstadt, Germany) and LDN193189 100 μM (04-0074; Stemgent, Cambridge, MA, USA) was added. The medium was subsequently changed every 48 h. Between DIV 4 and DIV 9, CHIR99021 3 μM (130-103-926; Miltenyi, North Rhine-Westphalia, Germany), SU5402 3 μM (SML0443; Merck, Darmstadt, Germany), and DAPT 10 μM (D5942; Merck, Darmstadt, Germany) were added, and KSR medium was gradually switched to N2 medium (neurobasal medium (21103049; Thermo Fisher Scientific, Waltham, MA, USA), N2 (17502-048; Thermo Fisher Scientific, Waltham, MA, USA), penicillin/streptomycin, NEAAs, and L-glutamine 2 mM). On DIV 10, 100% N2 medium plus recombinant human brain-derived neurotrophic factor 10 ng/mL (BDNF 450-02; Peprotech, Cranbury, NJ, USA), glial-derived neurotrophic factor 10 ng/mL (GDNF 450-10; Peprotech, Cranbury, NJ, USA), nerve growth factor 10 ng/mL (NGF N6009; Merck); neurotrophin-3 10 ng/mL (NT-3 450-03; Peprotech, Cranbury, NJ, USA), and ascorbic acid 200 μM (AA 49752; Merck, Darmstadt, Germany) was added. On day 16, organoids were plated into Matrigel-coated 24-well plates with and without glass cover, directly onto cultures of keratinocytes or in microfluidic devices, and maintained in N2 medium supplemented with the maturation factors or in different combinations of media where specified. Half of the appropriate medium was replaced every 72 h until samples were collected for analysis or for up to 90 days. Two fluorodeoxyuridine (FUDR) treatments were performed from DIV 17 to 26.

**Microfluidic culture system****.** The microfluidic compartmentalized bioreactor was previously designed [20], featuring two symmetric lateral chambers (43 μm high, 1.5 mm wide, and 7 mm long), one for the neuronal culture and the other one for the epidermal culture. Each compartment is connected to two wells at its ends, serving as culture medium reservoirs during cell culture. Additionally, the neuronal compartment is provided with two smaller reservoirs dedicated to hosting the soma of DRGOs. The two chambers are fluidically connected by 250 microgrooves, with a width of 5 µm, a height of 6 µm, and a length ranging from 830 to 930 µm. Microgrooves were conceived to guide axon growth, while preventing convective flow between compartments. Microfluidic devices were fabricated through soft lithography of PDMS on master molds. Briefly, the chamber design, except for the microgrooves, was printed on a high-resolution photomask (64,000 dpi). The master molds were fabricated in a cleanroom environment through a two-step photolithography procedure of SU-8 (Microchem, Round Rock, TX, USA) on silicon wafers. First, the microgroove pattern was created by direct laser writing of a layer of SU-8 2005 with a thickness of 6 μm, by using a maskless laser writer (MLA100, Heidelberg, Germany). Subsequently, a 43 μm thick layer of SU-8 2035 was spin-coated onto the wafer and exposed to a collimated UV beam through the photomask containing the features of the chamber. The PDMS layers were then fabricated by replica molding. PDMS (SYLGARD 184 Silicone Elastomer Kit, Dow, Midland, MI, Statele Unite) was poured on silicon wafers at a pre-polymer to curing agent mixing ratio of 10:1 (*w*/*w*) and cured at 65 °C for 3 h. After the curing phase, PDMS was peeled off the molds and trimmed, and through-holes were punched to obtain the lateral reservoirs (8 mm diameter) and the DRGO seeding sites (2 mm diameter). Finally, after plasma activation (Harrick Plasma), the PDMS layers were bonded to 24 × 50 mm microscope glass slides (Superslip Cover Glass, TED PELLA, Redding, CA, USA). The compartments were coated with Matrigel 1:200 (Thermo Fisher Scientific, Waltham, MA, USA) and incubated overnight at 4 °C. Each compartment holds about 150 μL of medium. Two 16-day-old DRGOs were seeded into the dedicated seeding sites, and 5 × 10^4^ NHEK cells were seeded in the other compartment and cocultured for 21 days before starting any analysis or treatment (Figure 1A). At 18 days of coculture, new NHEK cells were seeded as described above. Neuron and keratinocyte cultures were maintained as described above.

**Transcriptomic analysis****.** Bulk RNA-seq fastq files were mapped to the hg38 human reference genome with the Bowtie2 aligner. Differential gene expression and functional enrichment analyses were performed with DESeq2 and GSEA, respectively. Statistical and downstream bioinformatics analyses were performed within the R environment. Gene expression heatmaps were produced with GENE-E (Broad Institute). Data were deposited in the NCBI Gene Expression Omnibus repository with the GSE133755 GEO ID. RNA-seq datasets for human somatic dorsal root ganglia (SRA SRP077657) and human iPSCs (GEO GSE120081) were obtained from the NCBI GEO/SRA repositories for data mining. Gene ontology aggregated categories were produced by combining multiple GO datasets: heparan sulfate biosynthetic pathways (GO Heparan Sulphate Proteoglycan Binding; GO Heparan Sulphate Proteoglycan Biosynthetic Process, GO Heparan Sulphate Proteoglycan Biosynthetic Process Enzymatic Modification, GO Heparan Sulphate Proteoglycan Biosynthetic Process Polysaccharide chain B Biosynthetic process, GO Heparan Sulphate Proteoglycan Metabolic Process), integrin signaling genes (GO Integrin Mediated Signaling Pathway). ScRNA-seq fastq files were aligned to the GRCh38 human reference genome with cellranger count (10× Genomics) with default parameters and human Gencode v33 annotations (PMID:30357393). Gene count matrix log-normalization (10,000 scale factor), gene clustering, dimension reduction analysis (UMAP), differential gene expression analysis, and plotting were performed with Seurat within the R environment (R Core Team (2020). R: A language and environment for statistical computing. R Foundation for Statistical Computing, Vienna, Austria. URL: https://www.R-project.org/ (accessed on 29 April 2020)). Functional enrichment of gene ontology biological process datasets was performed with DAVID.

**Viruses****.** The laboratory strain HSV-1 HF (VR-260; ATCC) and a recombinant fluorescent virus kindly provided by Dr. P. Kinchington from the University of Pittsburgh were used. To follow the lytic and latent phases of HSV infection, a recombinant HSV-1 KOS (VR-1493; ATCC) that has been previously used and characterized for growth in other human neuron-based platform systems [12,22] was used in the present study. This virus was constructed in a manner detailed as previously described [23,24]. Briefly, it contains an insertion, at a unique NheI site in the N terminal region of the glycoprotein C (gC) open reading frame (ORF), a tandem dual reporter cassette so that the gC promoter drives expression of the monomeric red fluorescent protein (RFP) followed by a bGH polyadenylation site. This is then followed by a 1.5 Kb fragment of the HSV DNA containing the viral promoter for ICP0, followed by the enhanced green fluorescent protein (EGFP). The insertion site at the beginning of the gC ORF in the gC locus disrupts the gC open reading frame. LAT is the only actively transcribed gene from the virus in latently infected cells, and the boundary between transcription-permissive and -nonpermissive regions is located near the 3′ end of the ICP0. Thus, the ICP0 promoter is actively transcribed during latency, whereas the surrounding regions, such as the lytic-specific gene ICP0, are silenced, or there is a posttranscriptional constraint on the expression of ICP0 protein during reactivation from latency mediated by LAT.

**Infection of SH-SY5Y cells****.** SH-SY5Y cells were seeded on Matrigel-coated slides with a removable 12-well silicone chamber (Ibidi, Gräfelfing, Germany) at 10^4^/well. After 24 h, the cells were preincubated with ACV 100 μM in complete DMEM supplemented with 2% FBS. After another 24 h, cells were infected with HSV-1 HF (range: 0.01–0.0001 multiplicity of infection (MOI)) or KOS (0.0001 MOI) for 30 min at 37 °C and, after two PBS 1x washes to remove cell-free viral particles, were cultured with complete DMEM supplemented with 2% FBS and ACV 100 μM. After 5 days, ACV was removed, and cells were maintained in complete DMEM with 0.5% FBS to monitor spontaneous reactivations. At 7 (HSV-1 HF) or 25 (HSV-1 KOS) days post-infection (DPI), reactivation was induced using thermal stress (1 h 30 min at 43 °C), and cells were observed until reactivation occurred. Twenty-four hours after reactivation, cells were stained with Hoechst 33258 (Merck, Darmstadt, Germany) and fixed with PFA/PBS 2% or collected for total RNA extraction. Each condition was tested in quadruplicate. The fluorescence was measured by calculating mean gray values (average gray values within selected areas) using ImageJ software (NIH). Integrated density (area × mean gray value) was used to compare fluorescence intensities.

**Infection of DRGOs****.** For lytic infections, cell-free virus was adsorbed onto DRGOs seeded on Matrigel-coated slides. After 2 h, the cells were washed and cultured for 24 h, and live images were acquired. For latent infection, DRGOs were preincubated with ACV 100 μM in complete medium. After 24 h, DRGOs were infected with HSV-1 HF (range: 0.1–0.0001 MOI) or KOS (0.0001 MOI) and, after two PBS 1x washes, cultured with complete medium with ACV 100 μM. After 5 days, ACV was removed, and DRGOs were monitored for spontaneous reactivations. At 18 (HSV-1 HF) or 25 (HSV-1 KOS) DPI, reactivation was induced using thermal stress (1 h 30 min at 43 °C), and cells were observed until reactivation occurred. Twenty-four hours after reactivation, cells were stained with Hoechst 33258 (Merck, Darmstadt, Germany) and fixed with PFA/PBS 2% or collected for total RNA extraction. Each condition was tested in quadruplicate and integrated density was used to compare fluorescence intensities. 

**Infection of microfluidic chip**. Both NHEK and DRGOs were preincubated with ACV 100 μM for two days. Cells were infected with HSV-1 KOS (0.0001 MOI), and after 30 min of adsorption, cells were washed twice with PBS 1x to remove cell-free viral particles and cultured with the appropriate medium. Two experimental settings were established, using 6 microfluidic chips for each condition (corresponding to 12 DRGOs) plus controls. For experiment #1, only DRGOs were infected, and ACV 100 μM was added immediately after virus adsorption. For experiment #2, only NHEK cells were infected, and ACV 100 μM was added 6 h after adsorption. After 5 days, ACV was removed, and reactivation was induced using thermal stress (1 h 30 min at 43 °C). Two microfluidic chips (one for each experiment) were not subjected to thermal treatment as a spontaneous reactivation control. Live images were acquired when reactivation occurred, and 24 h later cells were collected for total RNA extraction. Integrated density was used to compare fluorescence intensities. Supernatants were collected from both chip chambers at different time points (1 h before infection; 1 h post-infection; before reactivation, 5 DPI; end of the experiment, 8 DPI) for testing for virus presence on Vero E6 cells.

**Phenotypic assay for the evaluation of ACV sensitivity**. First, 100 PFU/mL of virus was applied to Vero E6 monolayers. After 1 h, the virus mixture was replaced with complete DMEM with 2% FBS, 1% agarose (BD), and serially diluted preparations of ACV (range: 0.75–400 μM). Plates were incubated for 46 h. Cells were then fixed and stained, and lysis plaques were counted. IC_50_ values were calculated from the dose–response curve by linear regression analysis.

**Cell viability analysis****.** DRGOs were cocultured with HaCaT or NHEK cells on Matrigel-coated slides, using different culture media: complete DRGO media and complete DMEM with 10% FBS (50% *v*/*v*), complete DMEM with 10% FBS and neurotrophins, complete DRGO media and keratinocyte growth medium (50% *v*/*v*), and keratinocyte growth medium and neurotrophins. After 4 days, cytotoxicity was measured using an XTT-based in vitro toxicology assay kit (TOX2; Merck, Darmstadt, Germany) according to the manufacturer’s protocol. The incubation medium was collected after 4 h and read spectrophotometrically at a wavelength of 450 nm.

**Passive virus diffusion evaluation****.** Chips were prepared by seeding NHEKs in one chamber and Vero E6 cells in the other as “sensor” cells. Then HSV-1 KOS (0.01 MOI) was added to the NHEK chamber for 30 min for virus adsorption, and then both NHEK and Vero E6 chambers were washed twice with PBS 1x to remove cell-free viral particles, and the proper medium was added. In some chips, we added ACV 100 μM only in the Vero E6 chamber, testing two different protocols: pre- and post-infection, or only after virus adsorption. The treatment was included to assess whether, in the case of passive diffusion, it was still possible to protect the cells in the non-infected chamber. Supernatants from both chip chambers were collected at different time points (before infection, post-infection, 1 h post-infection, 24 h post-infection) and subsequently were applied to other Vero E6 cells seeded 24 h before on Matrigel-coated 96-well microplates (µClear, Greiner Bio-One, Kremsmünster, Austria). Plates were centrifuged at 2000× *g* for 15 min and incubated for 2 h. Then, after two PBS 1x washes, supernatants were replaced with complete DMEM with 2% FBS and plates were incubated for 72 h. Live images were acquired after 48 h. Hoechst 33258 was used for nuclear staining before cells were fixed with PFA/PBS 2%, 72 h post-infection (PI). The same evaluation was performed to monitor virus passive diffusion during latency experiments #1 and #2: supernatants from both chip chambers were collected at different time points (1 h before infection; 1 h post-infection; before reactivation, 5 DPI; end of the experiment, 8 DPI) to be tested on Vero E6 monolayers as described.

**Immunofluorescence of DRGO–NHEK connections****.** For immunocytochemical analysis, microfluidic chips were filled for 40 min at room temperature in 4% paraformaldehyde in PBS, washed with PBS at room temperature, permeabilized for 30 min in PBS containing 0.1% Triton X-100 and 10% normal goat serum, and incubated overnight at 4 °C in PBS containing 10% normal goat serum and primary antibodies: anti-cytokeratin 14 (K14) antibody (SP53; Merck, Darmstadt, Germany), anti-neurofilament heavy polypeptide (NF200) antibody (ab4680; Abcam, Cambridge, UK), anti-synapsyn 1 antibody (106001; Synaptic Systems, Göttingen, Germany), and anti-VGluT1 antibody (ab77822; Abcam, Cambridge, UK). Then, cells were washed three times with PBS and incubated for 2 h at room temperature with secondary antibodies. 

**Quantitative two-step reverse-transcription PCR (RT-qPCR).** Total RNA was extracted from pelleted cells with RNeasy Mini kit (74104; QIAgen, Hilden, Germany) according to the manufacturer’s instructions. First-strand cDNA synthesis was performed on equivalent amounts of RNA from each sample, using an oligo(dT)15 primer and SuperScript IV Reverse Transcriptase (18091050; Thermo Fisher Scientific, Waltham, MA, USA). Because the major species of latency-associated transcripts (LAT) is not polyadenylated, reverse transcription of LAT was achieved by using a LAT-specific primer, ICP0-3′ (Table 1). PCR was performed on equivalent amounts of cDNA; each reaction mixture contained 2 U Platinum Taq DNA polymerase (10966034; Thermo Fisher Scientific, Waltham, MA, USA), 0.2 μM each PCR primer, 0.2 mM each deoxyribonucleotide triphosphate and PCR buffer, and 1.5 mM MgCl_2_, and each reaction was performed in a MultiGene OptiMax (Labnet International, Cary, NC, USA) thermal cycler with 40 cycles of 94 °C (30 s); 62 °C (BRN3A, NAV1.7, ACTB), 49 °C (LAT), 51 °C (gG), 55 °C (GAPDH), 59 °C (gB), or 61 °C (gD) (30 s); and 72 °C (30 s). Oligonucleotide primers were obtained from Eurofins Genomics (Louisville, KY, USA) and are listed below. Densitometry of GelRed-stained agarose gels was performed with Image-Lab software 6.0.1 (Bio-Rad, Hercules, CA, USA). For each gene, the threshold at which the product was detected was compared with the endogenous (GAPDH) control. Levels of viral mRNAs were normalized to GAPDH RNA levels to correct for recovery because GAPDH levels at 6 hpi have been shown to be similar to levels in mock-infected cells [25].

**Statistical analysis****.** Two-way ANOVA and Sidak’s multiple comparisons test were performed for the evaluation of gene expression differences in monoculture results and to compare fluorescence intensities in all experiments, while Tukey’s multiple comparisons test was used for the analysis of data from experiments #1 and #2.

**Data availability.** Data generated during the study are available in the NCBI GEO public repository with accession Nos. GSE133755 (https://www.ncbi.nlm.nih.gov/geo/query/acc.cgi?acc=GSE133755 (accessed on 29 April 2020)) and GSE148212 (https://www.ncbi.nlm.nih.gov/geo/query/acc.cgi?acc=GSE148212 (accessed on 29 April 2020)).

## 3. Results

### 3.1. Setting the Neuronal–Epithelial Coculture Model in a Patterned Microfluidic Chip Culture System

A thorough analysis of virus reactivation requires a humanized system where keratinocytes and neurons can be simultaneously cultured for an extensive time to develop direct connectivity. To this aim, we decided to evaluate the possibility to develop a new neuron–keratinocyte coculture system using hiPSC-derived DRGOs that we have recently derived and showed to generate molecularly defined peripheral sensory neurons capable of making specific contacts with their respective target tissues [20]. DRGOs were obtained by differentiating transgene-free hiPSC-derived aggregates in 3D culture conditions and sequentially exposed to small molecules to induce peripheral nervous system identity (Appendix A). This protocol led to the generation of rounded spheroids emitting radially projecting axonal projections (Appendix A) which can be maintained in vitro for several weeks. Successful commitment to peripheral sensory neuronal identity was shown by high expression levels of specific molecular markers, including BRN3a and NAV1.7 (Appendix A). Triple immunostaining identified peripheral sensory neurons expressing different combinations of TRK receptors, confirming their correct differentiation into nociceptive (TRKA+/TRKC−), mechanoreceptive (TRKB+/TRKC+), and proprioceptive (TRKB−/TRKC+) neurons (Appendix A–F). Bulk and single-cell RNA-seq confirmed and extended this analysis, showing that DRGOs also include satellite cells and Schwann cell progenitors, mirroring the overall cell composition of somatic DRGs [20].

For establishing cocultures, either immortalized (HaCaT) or primary (NHEK) keratinocytes were seeded on coated slides, and 24 h later, DRGOs were added to the monolayers. Different culture media were tested to find the correct composition to meet the nutritional requirements of all the cell types. Unfortunately, all tested conditions were unable to ensure proper cell viability, as shown by their morphology, connectivity, and survival (Appendix A). This susceptibility to medium composition was confirmed by the cell proliferation assay (Appendix A). Thus, direct coculture between DRGOs and keratinocytes remained unfeasible due to inherently different culture condition requirements.

To overcome the direct coculture limitations, we sought to reconstitute the coculture system into an appropriate microfluidic chip to prevent media mixing, as already tried by others in similar contexts [12,15,16,17]. We used a custom-designed chip composed of two chambers, one for each cellular type, connected together by microgrooves to allow axonal projections to reach and connect with the keratinocytes (Figure 1A), which has already proven to be suitable for DRGO axonal studies [20]. Long-term cell survival and proper cell morphology were detected in both chambers. However, when considering neuron-to-target connections, DRGO axonal terminals were degenerate when they reached the HaCaT chamber (Figure 1B), suggesting that the serum necessary for their growth negatively interfered with normal neuronal behavior, similarly to what we observed in previous direct cocultures, and differently from what observed in literature with rat neurons under similar conditions [16]. On the contrary, immunofluorescence analysis confirmed the innervation of the keratinocyte with multiple DRGO axons and the formation of stable connections when DRGOs reach NHEKs cells. Indeed, we clearly observed synapsin- and vGlut-positive DRGO free nerve endings on NHEKs closely recapitulating the typical anatomical configuration of the nociceptive contacts within the skin keratinocytes in vivo (Figure 1C). These results confirmed that the design of the microfluidic chip with an array of microgrooves connecting the two lateral chambers was efficient in preventing culture media mixing, while enabling axonal connectivity between the two compartments. Additionally, we provided evidence that DRGOs are a suitable model for multiorgan studies in vitro, reconstituting stable and topographic neuronal circuitries with different peripheral tissue targets.

### 3.2. DRGO Gene Expression Competence to HSV-1 Infection

To evaluate the possibility to use DRGO sensory neurons as a target for HSV, we next evaluated the expression of the main HSV cellular receptors in DRGOs. To this end, we carried out a computational analysis on global and single-cell RNA-sequencing datasets on isolated DRGOs generated in our previous work [20]. When compared to iPSCs at the global transcriptomic level, we confirmed that DIV40 DRGOs and primary DRGs are closely related with respect to the expression of the main genes involved in virus–host interaction, in particular the main HSV-1 receptor NECTIN1, but also TNFRSF14 (HVEM), PILRA, NRP1, MYH9, ITGA5, ITGA8, ITGB6, and ITGB8 (Figure 2A). Interestingly, this similarity is not strikingly related to HSV-1 receptors but is a more general feature of DRGOs, underlined by the close relation to primary DRG in regard to heparan sulfate biosynthetic pathways and, more in general, integrins (Figure 2B,C). Remarkably, such similarities between primary and stem cell-derived ganglion neurons were evident even if DIV40 DRGO neurons were generally not completely mature yet [20], and some differences were still present, such as in Figure 2A. 

To better understand the cellular diversity of DRGO receptor composition in more mature DRGOs, we re-analyzed our previously published scRNA-seq datasets collected for 5363 single cells isolated from two independent batches of DIV80 DRGOs, from which we identified 14 cell clusters based on their overall gene signature, containing all three sensory neuron subtypes (nociceptive, mechanoreceptive, and proprioceptive neurons), satellite cells, and Schwann cells (Figure 3A), fully recapitulating the cellular composition of somatic DRGs. Interestingly, we observed different combinations of expressed HSV receptors in the different DRGO sensory neuronal and glial cells (Figure 3B–J). All neuronal populations (C3, C4, C8) exhibited NECTIN1 and NRP1 expression, while in glial cells (C9–C13) these receptors were almost absent. The nociceptor cluster (C4) presented, differently from proprioceptors, high levels of SDC1 and HS3ST3B1. The mechanoreceptor cluster (C8) was the only neuronal group expressing ITGB8 and, together with Schwann cells, ITGA6. In glial cells, the satellite cell clusters (C9–C11) were characterized by low levels of HS3ST3B1 and ITGA6, and Schwann cell clusters (C12, C13) present a high level of ITGA6 and low level of ITGB8 (Figure 3B–J). Collectively, these findings demonstrate that DRGOs resemble primary DRGs with respect to the global expression of HSV receptors and binding factors. Moreover, single-cell analysis revealed a marked cell-type specific expression pattern of known important components of the host-to-viral interaction, suggesting that each single sensory neuronal population might have a specific profile of its interaction with HSV, and DRGOs might represent a convenient system which recapitulates the full complexity of the DRG cellular system.

### 3.3. Validation of the Neuronal–Epithelial Microfluidic Platform for Viral Infectivity Studies

We, next, asked to what extent this topographic human neuronal–epithelial circuitry can be informative for studying the complex behavior of HSV infectivity. At first, we asked if virus particles were blocked in their passive diffusion between chambers, even during lytic infection when there is wide viral spreading among cells and massive production of new virions. This is of primary importance to validate our model for studies of neuron-to-cell HSV diffusion. We used a recombinant HSV-1 KOS in the present study, incorporating enhanced green fluorescent protein (EGFP) driven by the viral promoter ICP0 and monomeric red fluorescent protein (RFP) driven by the viral promoter glycoprotein C (gC) as reporters for infected cells or cells in lytic phase, respectively. This characteristic allowed us to follow the lytic and latent phases of HSV infection. Moreover, strains of HSV-1 have been noted to vary greatly in their virulence and reactivation efficiencies in animal models [29]. Indeed, strain KOS shows little to no reactivation in the mouse and rabbit models of induced reactivation because their genomes entering neurons are more prone to rapid heterochromatinization and this results in a reduced ability to reactivate from latency [30]. In our work, we prefer the KOS strain precisely to limit the number of uncontrolled reactivations [30,31,32,33,34,35].

We prepared chips seeded with NHEKs in a chamber and Vero E6 cells in the other as “sensor” cells. Then, only NHEK cells keratinocytes were infected with a hundred times more virus than would be used in subsequent experiments. Next, supernatants were collected at different time points (before infection, 1 h post-infection, 24 h post-infection) from both the NHEK chamber, to monitor the productive infection over time, and the Vero E6 chamber, to monitor the possible passive diffusion of virions between the two chip compartments, and finally assayed for infectious virus on other Vero E6 cells seeded on plates. In some chips, ACV 100 μM was added only in the Vero E6 chamber to test two different protocols: pre- and post-infection, or only after virus adsorption. The treatment was included to assess whether, in the case of passive diffusion, it was still possible to protect the cells in the non-infected chamber (Appendix A). Infectious virus was detectable in the supernatant of infected NHEK cultures, but it was never detected in the supernatant of the uninfected chamber. This validates our microfluidic model for the study of the neuron-to-cell virus spreading, as the cell-free virus cannot migrate between the different compartments of the chip through passive diffusion alone, as previously demonstrated for Texas Red dextran [36].

### 3.4. Successful Establishment of HSV-1 Latency in DRGO Neurons

As virus reactivation was a main objective of this study, a novel protocol for HSV-1 establishment of latency was set up using two virus strains: HSV-1 HF, a laboratory strain, and the recombinant HSV-1 KOS. The induction of HSV latency is performed by pre-treating cells with acyclovir (ACV) before virus adsorption. Therefore, a fundamental requirement for both HF and KOS viral strains is their sensitivity to ACV. To this aim, in vitro phenotypic assays were performed, and the results confirmed that the amount of antiviral to be used in the latency protocol would be well above the IC_50_ (Figure 4A). The HF non-recombinant strain was used to tune our experiments. SH-SY5Y neuroblastoma cells and DRGOs were infected with different virus concentrations after ACV pretreatment. After adsorption, cells were maintained in culture media supplemented with ACV for 5 days. SH-SY5Y neuroblastoma cells were then cultured without antiviral for 2 days to monitor for spontaneous reactivations. Then, thermal stress was used for inducing virus reactivation, and the cytopathic effect caused by productive infection was visible 2 days later. Reactivation thermal treatment was performed 5 days following DRGO ACV treatment, and about 10 days later the cytopathic effect on cells started to appear (Figure 4B). Noteworthily, virus reactivation from latency was successfully obtained even using three-dimensional cultures. Once the protocol was developed using a reference strain, the recombinant KOS virus was used. Twenty days after antiviral removal, we applied the thermal treatment to obtain detectable HSV1-KOS reactivation 3 days later. This dilation time after ACV removal from the medium was mandatory to ensure that what we observed was indeed a controlled and not a spontaneous reactivation. With this procedure, neuroblastoma cells show spontaneous reactivation in one out of eight total samples at 11 DPI. On the contrary, no spontaneous reactivation occurred in DRGOs.

Live imaging was obtained for latently infected cells in culture, and it was possible to detect the presence of latently infected cells before thermal treatment in both SH-SY5Y and DRGO cultures as highlighted by the EGFP signal (Figure 4C). Lytic infection could be visible only after thermal stress, monitored by the combination of both GFP and RFP fluorescence signals. Integrated density was used to compare both green and red fluorescence signals of reactivated and latently infected cells, highlighting the actual difference only between the RFP signal detected in both SH-SY5Y and DRGOs, even if it had statistical significance (*p* < 0.05) only when using organoids (Figure 4D).

To assess the differences in gene expression profiles during latency and reactivation, cells were harvested at the end of the experimental protocol for total RNA extraction. Quantitative two-step reverse-transcription PCR (RT-qPCR) showed a slight but not significant reduction in LAT transcripts 24 h post-reactivation in both SH-SY5Y cells and DRGOs, using both viruses (Figure 4E). Instead, an important increase in glycoprotein G (gG) expression was observed in both HSV-1 HF and KOS infected cells (*p* < 0.0001 and *p* < 0.01, respectively) and DRGOs (*p* < 0.0001 using both viruses). Next, we observed an increase in the late gene expression of both glycoprotein D (gD) and B (gB). It is interesting to note that during the latent phase, SH-SY5Y cells retain partial expression of lytic products, as well as lytic RFP fluorescence, while in DRGOs it is almost absent, suggesting a more physiological and sustainable viral latent phase in the human organoid model.

In summary, all data confirmed that we were able to establish stable and long-term HSV-1 latency in DRGOs, allowing us to better model virus reactivation in human sensory neurons using three-dimensional cultures directly differentiated from hiPSCs.

### 3.5. Experimental Settings for Latency and Reactivation in Microfluidic Culture System

Given the results described above, two experimental settings were tested for the study of HSV-1 reactivation from latency, namely the anterograde virus spread experiment (EXP #1), involving direct infection and latency of DRGOs (Figure 5A,B), and the retrograde–anterograde virus spread protocol (EXP #2) with the establishment of latency in DRGOs following NHEK infection (Figure 5C,D), summarizing the physiological behavior of HSV-1. For both experiments, NHEKs were seeded together with DRGOs to allow axonal contacts and seeded again 18 days later to fill the chamber (Appendix A). After 21 days, all chambers were pretreated with ACV and infected for 30 min with recombinant HSV-1 KOS before re-supplementing ACV, except for NHEK cells of EXP #2, which were treated with ACV 6 h post-adsorption to allow virus replication and spreading to DRGOs through axonal connections and receptor recognition. All cells were cultured with antiviral for 5 days. Then, ACV was removed, and thermal stress was used for inducing virus reactivation. The cytopathic effect caused by productive infection was visible 2 days later. Supernatants from different time points (1 h before infection; 1 h post-infection; before reactivation, 5 DPI; end of the experiment, 8 DPI) from both DRGO and NHEK chambers were tested on Vero E6 monolayers (Appendix A). We did not observe a clear signal 24 h after reactivation, so cell pellets were processed for RNA extraction to assess the expression of both HSV-1 latency transcripts and late genes. While infectious virions were detected in the supernatant of NHEK after virus adsorption (EXP #2), they were never detected in the supernatant of uninfected chambers, confirming that virions never passively transfer between chambers, making our neurosensory–epithelia connection on a chip a feasible tool for modeling HSV-1 reactivation processes in vitro.

#### 3.5.1. Anterograde Virus Spread in Neuronal–Epithelial Culture System

The first experiment involved direct infection of DRGOs with low-titer HSV-1 and the establishment of latency by an ACV-supplemented medium (Figure 5A). Gene expression profiles were assessed, and exclusively LAT transcripts were detected in DRGOs only, confirming once again both latency protocol efficacy and no passive virus diffusion occurring through the NHEK chamber (Figure 6A). When controlled reactivation occurred following ACV removal and thermal stress, the virus spread to NHEKs through axonal connections to keratinocytes by anterograde virus spread (Figure 5B). Analysis of gene expression revealed that LAT transcripts were not significantly decreased 24 h post-reactivation in DRGOs, as already found in monoculture data (Figure 6B). They were different in NHEK cells (*p* < 0.01), as no transcripts were detected in that chamber during latency. No expression of late genes (gG, gB, gD) was detected during latency in both DRGOs and NHEKs, but they were actively transcribed during reactivation in both cell chambers, as shown by graphs (DRGOs: gG *p* < 0.001, gB *p* < 0.0001, gD *p* < 0.0001; NHEK: gG *p* < 0.0001, gB *p* < 0.05, gD *p* < 0.0001).

Live imaging of the two cell chambers was concordant with gene transcript analysis (Figure 6C). Latency was detected by the EGFP signal only in DRGOs. After the thermal treatment, the productive infection was detected in both the neuron and keratinocyte chambers thanks to the co-presence of EGFP and RFP fluorescence. Values of fluorescence intensity confirmed what was observed: the lytic signal was detected only during productive infection in both cell populations (*p* < 0.0001), while the green signal was present in NHEK images only after the induction stimuli (*p* < 0.0001). DRGOs showed EGFP fluorescence in both latent and reactivate states, albeit with a slight but significant difference (*p* < 0.05) (Figure 6D). No signal was detected in uninfected microfluidic chips, confirming that what was observed was effectively due to the virus reactivation process.

#### 3.5.2. Validation of the Retrograde–Anterograde Virus Spread 

The EXP #2 is designed to test the retrograde–anterograde virus spread. Gene expression profiles confirmed that our model fits with the literature data on latency established only in neurons and not in non-neuronal cells, as no virus transcripts other than LATs were detected in DRGOs and none were detected in keratinocytes (Figure 7A). Then, ACV removal and thermal stress drove HSV-1 reactivation and neuron-to-cell spreading, leading to active replication in keratinocytes (Figure 5D). Once again, LAT transcripts were not significantly decreased 24 h post-reactivation in DRGOs, while they were actively transcribed in NHEKs only after reactivation (*p* < 0.0001) (Figure 7B). Late genes (gG, gB, gD) were actively transcribed only during reactivation in both chip compartments, and overall lower levels of expression were observed in EXP #2 than #1 for both DRGOs and NHEKs, as shown by qPCR analysis (DRGOs: gG *p* < 0.0001, gB *p* < 0.01, gD *p* < 0.001; NHEKs: gG *p* < 0.0001, gB *p* < 0.001, gD *p* < 0.0001).

Living cell imaging confirmed the gene expression results (Figure 7C). In fact, EGFP signal was detected only in DRGOs, corroborating latency establishment exclusively into neurons. As expected, fluorescence was less intense than EXP #1 due to the indirect modality of neuronal infection. After the thermal treatment, the productive infection was detected in both cell chambers. These data are confirmed by immunofluorescence signal measurements: RFP signal was detected only during productive infection in both cell populations as for EXP #1 (*p* < 0.0001), and EGFP signal was detected in NHEK images only after the induction stimuli, but less intensely than in the previous experiment (*p* < 0.001). DRGOs showed an important green signal as well in both latent and reactivation states, but this time without a significant increase (Figure 7D).

In this case as well, no fluorescence signal was detected in microfluidic chips with uninfected cells, validating live imaging observations. These results showed that DRGOs were successfully infected by HSV-1 through retrograde axonal transport from infected keratinocytes or cell-free virus, as not every axon contacts NHEKs in a coculture system, and thermal treatment led to direct neuron-to-cell spreading of infection. Microfluidic chips were maintained in the latent state as spontaneous reactivation control until the end of the experiment (8 DPI). They were not subjected to thermal stress after ACV removal from the culture medium, and cells were collected to test gene expression. As expected, only LAT transcripts were detected in DRGOs, while no virus gene expression was detected in keratinocytes (Appendix A).

Collectively, the presented data validate the establishment of a new human sensory neuron-to-keratinocyte circuit on a chip suitable for fully recapitulating the key HSV-1 viral processes of reactivation from latency.

## 4. Discussion

To fully dissect the neurotropic viral reactivation, a suitable experimental in vitro system of human origin that recapitulates the cellular complexity and stereotyped connectivity present in vivo is necessary. In this regard, several cellular settings have been tested in cell culture or coculture, using either human and non-human-derived primary or immortalized cells. However, most of these systems are not adaptable to high-throughput applications involving post-mortem human or rodent tissues. Others rely on hiPSC derivatives that mostly resemble CNS neurons instead of PNS neurons. The direct consequence of this is the absence of a reliable fully human-based model to understand viral processes in greater depth and faithfully test antiviral drug candidates. Most preclinical studies evaluate HSV vaccine efficiencies in mouse models. Unfortunately, in contrast to humans, recurrent HSV shedding and recurrent herpetic disease do not occur in mice because HSV spontaneous reactivation is either extremely rare or does not occur in mice [37]. Therefore, although mouse studies have provided ample crucial information regarding immune response against primary HSV-1 and HSV-2 infections [38,39,40,41,42], the efficacy of human epitope-based therapeutic vaccines against recurrent shedding and disease cannot be assessed in mice. The lack of an accurate animal model that translates human immune responses and diseases certainly limits the proper preclinical assessment [43].

Here, we described and validated a novel in vitro human 3D cellular system on a chip that could fill this current gap. The human neuronal component relies on the use of DRGOs, DRG-like organoids derived by 3D differentiation of hiPSCs into the plethora of sensory neurons and glial cells, and is capable of directly connecting with keratinocytes. These findings confirm and further extend the intrinsic capacity of DRGO sensory neurons to contact and make stable connections in vitro with the authentic peripheral target tissues of DRGs in the human body, as we previously showed for the reconstitution of the muscle spindles together with the intrafusal muscle fibers [20]. Moreover, our microfluidic chip, developed for 3D organoids, is confirmed to be a reliable tool for miniaturization studies on subcellular components and provides a great improvement to current compartmentalized neuronal systems. Thus, this human neuron–skin circuit on a chip represents an invaluable platform for studying viral mechanisms and kinetics in a 3D patterned circuit fundamentally similar to its in vivo counterpart and enables biochemical and molecular analyses that would be difficult, if not impossible, to perform in vivo. 

Different from primary rat neurons [16], the serum present in the HaCaT medium and the low CaCl_2_ concentration in the NHEK medium [44] could be the possible causes of the detrimental axonal effect observed on DRGO axons. This human-specific serum sensitivity of our 3D organoids underlines the difference between human and rodent models in regard to cellular behavior, and the need for a human-specific cellular model to recapitulate in vitro human biological processes, such as HSV-1 infection.

Then, the possibility of having a pathogen-permissive neuronal model able to be correctly recognized from the virus confirmed the feasibility of DRGOs as a suitable model for in vitro studies on a human genetic and infective disease affecting peripheral sensory neurons and allowed us to set up neuron-to-skin connectivity, suitable for multiple studies, including HSV-1 tropism. We showed that the pathogen competence of DRGOs is due to the presence of many specific HSV receptors, confirmed by the global transcriptome profile, similar to what was observed in primary ganglia. Noteworthily, through the mining of scRNA-seq datasets, we revealed that DRGO cell types express a unique combination of HSV receptors. These surprising findings might suggest that the kinetics and efficacy of HSV infection are not identical among the different DRG neuronal populations providing a different impact on the overall viral pathological effects. Future studies will be necessary to understand the transcriptional cascade which links a particular pattern of expressed viral receptors with the specific identity of the sensory neuronal cell type, since now we lack single-cell RNA data from primary human DRGs.

Primary infection using low-titer virus and cell medium supplemented with ACV at least 35 times above the IC_50_ values obtained through sensitivity assays led to latency establishment in cells, and thermal stress led subsequently to controlled reactivation of infected cell culture, as performed with other HSV-1 strains. Gene expression profiles were assessed to confirm the two stages of infection by analyzing LAT and late gene transcription. The use of the recombinant HSV-1 virus greatly facilitated the subsequent experimental procedures, as it allowed live imaging discrimination of cells in a latent state or reactivated. Its reporter genes (EGFP and RFP) are under the control of the viral promoter ICP0, which is actively transcribed during latency [45,46,47], and the lytic viral promoter glycoprotein C, respectively. Therefore, latently infected cells are easily identified due to the green fluorescent signal, which turns yellow if there is a lytic infection following reactivation. 

Our system provides a reliable model for studying HSV-1 reactivation from latency in neurons and subsequent infection of keratinocytes through neuron-to-cell connectivity. Both live-cell imaging and gene expression profiling confirmed the achievement of this result, as only green fluorescence and LAT transcripts were detected in DRGOs. Conversely, NHEKs remained virus-free as demonstrated by the lack of passive diffusion of infective virus particles. Then, thermal stress led to virus reactivation in neurons, and immunofluorescence and gene expression analysis demonstrated productive infection in both chip chambers. As we confirmed that no passive diffusion is possible within microfluidic compartments, HSV-1 reactivation from latently infected DRGOs led to NHEK primary infection only through axonal contacts.

The positive results obtained with this first experiment allowed us to establish a system to better model the physio-pathological conditions of HSV-1 infection, the retrograde–anterograde virus spread. In this experimental setting, NHEKs were primarily infected without ACV to facilitate virus replication and spreading through retrograde transport to DRGOs. Conversely, subsequent supplementation with ACV blocked virus productive infection into the keratinocytes, concomitantly allowing the establishment of latency in neurons as confirmed by both live-cell imaging and virus transcript analysis. The lower intensity compared to the EXP #1 conditions is probably due to the greater complexity of the model that could lead to fewer DRGO neurons infected due to retrograde migration compared to direct infection. In fact, in the first experiment, the virus adsorbed on the whole organoid, whereas in the retrograde infection, the latency is established only by the portion of viruses undergoing retrograde trans-axonal migration. No lytic or latent infection was detected in the NHEK chamber. Supernatants tested on sensor cells corroborated the presence of infective virus particles only in the NHEK chamber after adsorption, but not in the DRGO compartment or in both compartments in the subsequent time points. Thus, we successfully obtained latency in DRGO neurons from HSV-1 replication into keratinocytes and the subsequent virus spreading through axonal contacts and retrograde transport, closing our retrograde–anterograde virus spread in the microfluidic platform. Interestingly, transcriptional analysis from NHEKs showed an imbalance towards early gene expression, suggesting a possible lower virus titer able to reach keratinocytes or a delayed timing of anterograde transport compared to what resulted from the simplified method described above.

The results shown confirmed that our neuron–skin connectivity is a reliable platform for the molecular-level study of HSV axonal transmission during reactivation from latency in a topographic coculture system. Its fully human composition and the miniaturization and standardization, made possible by microfluidics technology, provide this device with unique features, perfectly fitted for applications spanning from the fine dissection of molecular mechanisms occurring during neurotropic virus infection to the translational application for both therapeutic and diagnostic purposes.

## 5. Conclusions

The development of stable neuron-to-keratinocyte connectivity in a microfluidic device enables the characterization of molecular mechanisms involved in the reactivations and spreading of neurotropic viruses in a fully human context. Next, our new knowledge of the viral receptors expressed by human neurons will facilitate the mechanistic dissection of the complex virus–host interactions. This on-chip platform has also an important application for the validation of HSV infection biomarkers, treatment responses, or potential curative therapeutic interventions, which cannot be studied using other in vitro or even in vivo systems. Furthermore, our new model was validated by using a well-characterized recombinant virus. Therefore, new studies using also clinical virus isolates will be important for shedding light on unknown pathophysiological aspects of herpesvirus reactivation. Finally, the use of organoids with defined cellular populations will enable the development of drug screening platforms. This opens the door to further expanding this culture platform to other neurotropic viruses to dissect reactivation mechanisms and to identify novel targets for antiviral therapy.

## Figures and Tables

**Figure 1 biomedicines-10-02068-f001:**
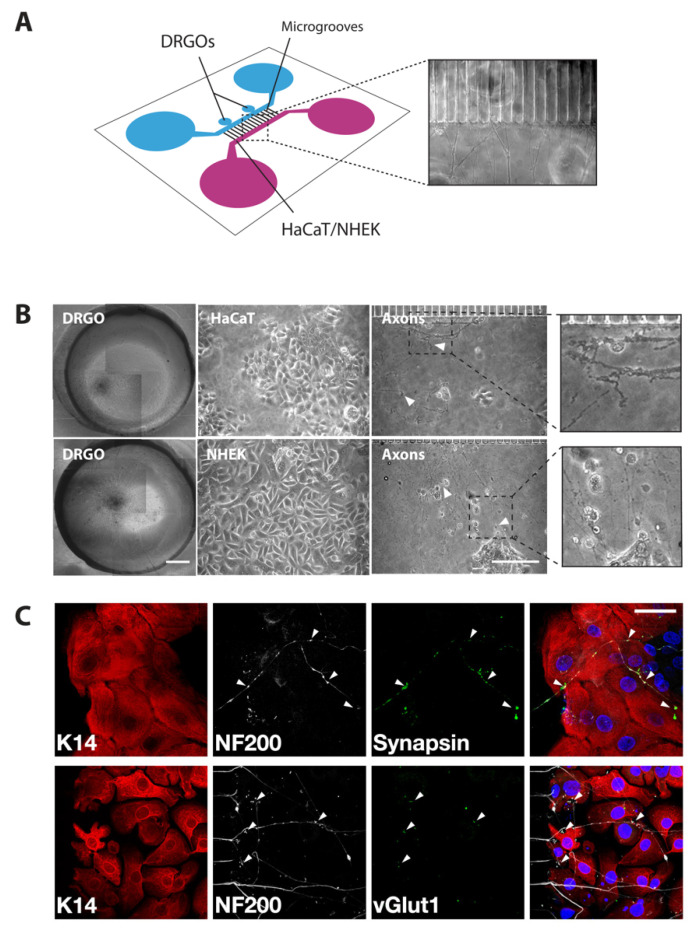
Coculture of DRGOs and keratinocytes on a chip. (**A**) Graphical representation of microfluidics chip and bright-field image (20× magnification) of axonal extensions from DRGO chamber to NHEK chamber through microgrooves. (**B**) Bright-field microscopy images of cells cocultured in microfluidic chips, with neuron-to-cell connection (white triangles). Scale bars: DRGO, 200 μm; HaCaT/NHEK/axons, 100 μm. (**C**) Immunofluorescence analysis of free nerve ending neuron-to-cell connection (arrowheads) using K14 (red), NF200 (white), synapsin (green, up), and vGlut1 (green, down). Total nuclear DNA is counterstained with Hoechst (blue). Scale bar, 50 μm.

**Figure 2 biomedicines-10-02068-f002:**
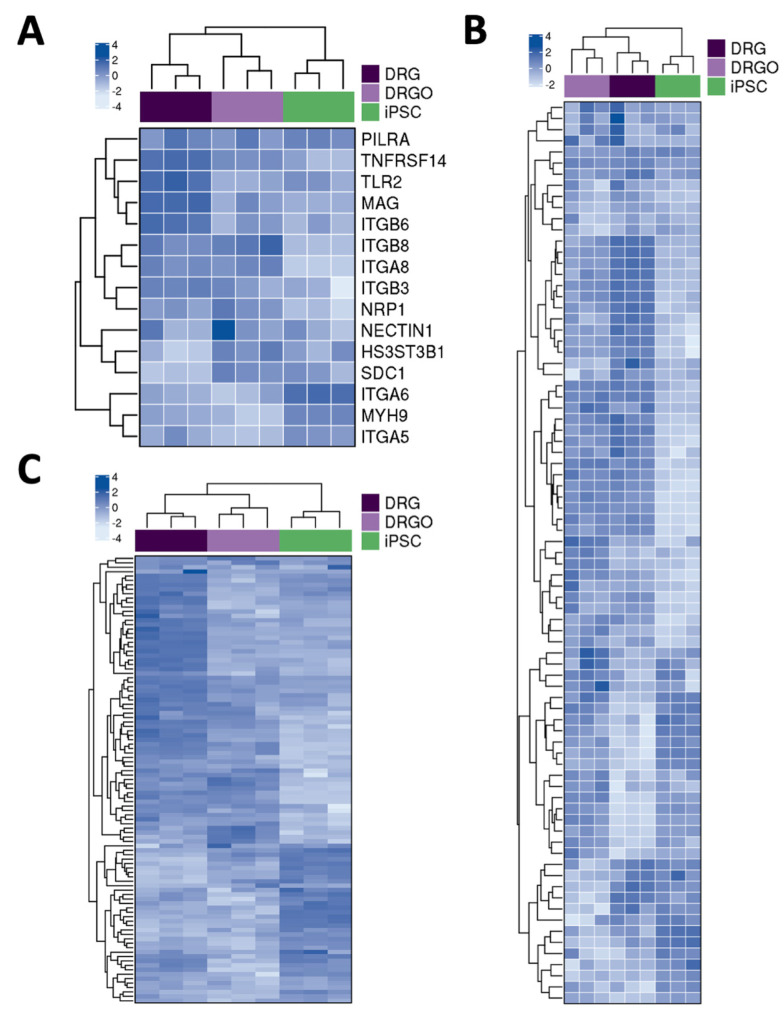
Global gene expression profile of HSV receptors in DRGO. (**A**) Supervised gene expression heatmaps showing the HSV receptor genes differentially expressed either in DRG vs. iPSC or in DIV 40 DRGO vs. iPSC; the correlation between samples is also shown as an unsupervised hierarchical clustered dendrogram on the side. (**B**,**C**) Gene expression heatmaps showing the differentially expressed genes belonging to aggregated GO categories associated with heparan sulfate biosynthetic pathways (**B**) and integrin-mediated signaling pathway (**C**); the correlation between samples is also shown as an unsupervised hierarchical clustered dendrogram on the side.

**Figure 3 biomedicines-10-02068-f003:**
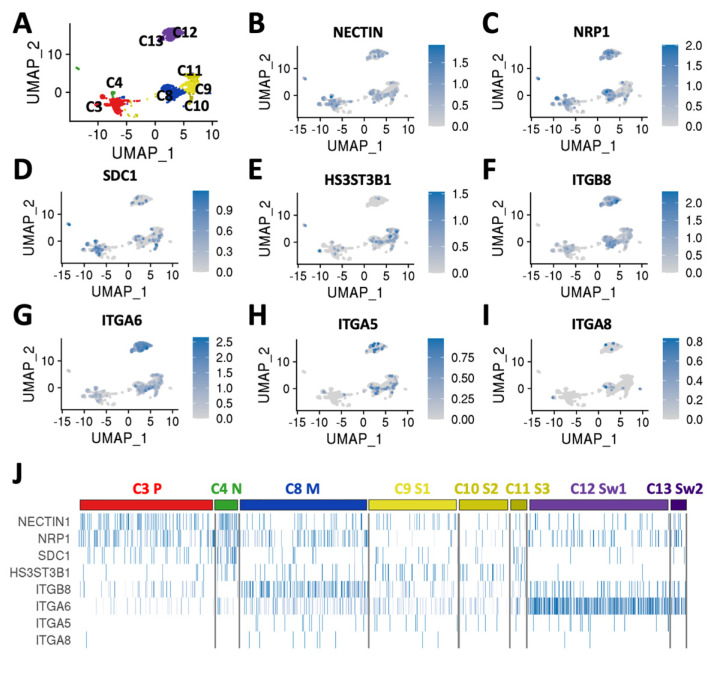
Single-cell transcriptional profile of HSV receptors in DRGO. (**A**) Uniform Manifold Approximation and Projection (UMAP) plot displaying multidimensional reduction and clustering of single-cell RNA-seq data from DIV 80 DRGOs showing clusters of mature sensory neurons (proprioceptors C3, nociceptors C4, mechanoreceptors C8), satellite cells C9–C11, and Schwann cells C12 and C13 (modified from [20]). (**B**–**I**) UMAP plots highlighting normalized expression values of NECTIN1 (**B**), NRP1 (**C**), SDC1 (**D**), HS3ST3B1 (**E**), ITGB8 (**F**), ITGA6 (**G**), ITGA5 (**H**), and ITGA8 (**I**). (**J**) Heatmap showing normalized expression values of cell lineage-specific genes within the different clusters.

**Figure 4 biomedicines-10-02068-f004:**
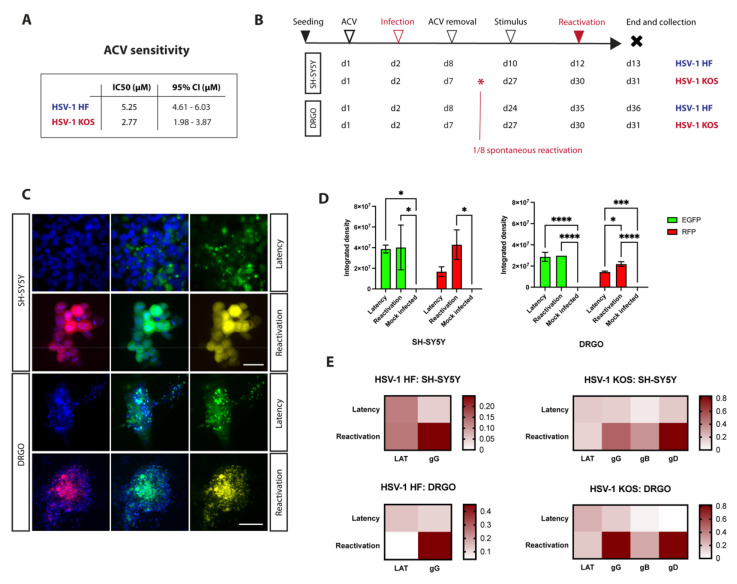
HSV-1 latency and reactivation protocol setup. (**A**) ACV phenotypic assay performed on HSV-1 laboratory strain (HF) and recombinant virus (KOS). (**B**) Schematic representation of different protocol timing tested on SH-SY5Y and DRGOs, using both virus strains. (**C**) Immunofluorescence analysis of SH-SY5Y and DRGOs during virus latency and reactivation. Virus protein gC (red, first column) is transcribed only during productive infection, while ICP0 promoter (green, second column) is active during both stages of infection. Total nuclear DNA is counterstained with Hoechst (blue); the last column shows the merge of green and red signals. Scale bars: SH-SY5Y, 30 μm; DRGO, 200 μm. (**D**) Levels of ICP0 (green fluorescent signal) and gC (red fluorescent signal) expression measured as integrated density values for SH-SY5Y and DRGOs during latency and reactivation. Mean ± SD is reported, * *p* < 0.05, *** *p* < 0.001, **** *p* < 0.0001. (**E**) Heatmap showing virus gene expression analysis of SH-SY5Y and DRGOs during latency and reactivation using both HSV-1 strains (dark red = high expression; white = no expression). Each condition was tested in quadruplicate.

**Figure 5 biomedicines-10-02068-f005:**
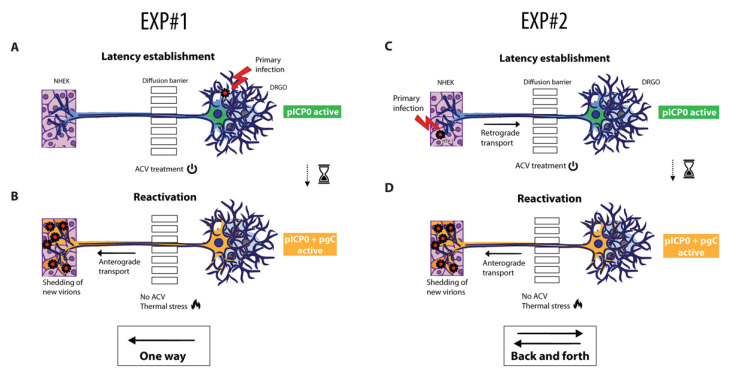
Testing HSV-1 latency in the microfluidic culture system. Graphical representation for HSV-1 latency establishment directly in DRGOs following anterograde virus spread (EXP #1, (**A**,**B**)) or retrograde–anterograde virus spread (EXP #2, (**C**,**D**)). In EXP #1, organoids were infected (**A**) and latency was obtained through ACV addition to the culture medium. Thermal stress (**B**) led to controlled reactivation, and anterograde transport of virus particles resulted in NHEK lytic infection. EXP #2 instead was carried out by HSV-1 latency establishment indirectly in DRGOs (**C**). NHEK cells were infected, and retrograde transport of virus particles resulted in HSV-1 latency establishment in organoids. Then, thermal stress (**D**) led to controlled virus reactivation, and anterograde transport of virus particles resulted in NHEK lytic infection.

**Figure 6 biomedicines-10-02068-f006:**
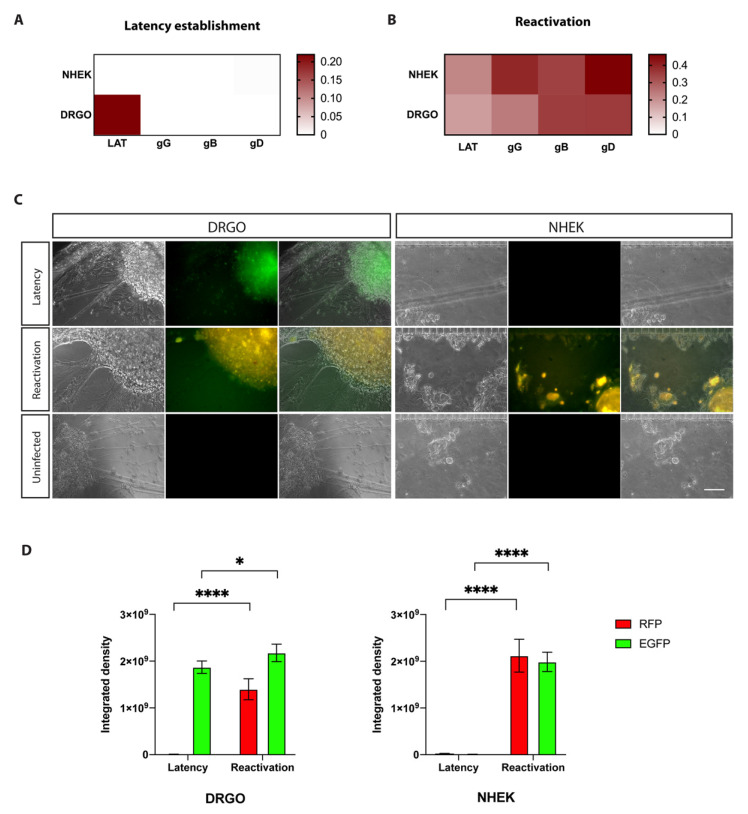
Anterograde virus spread experiment (EXP #1). Virus gene expression profiling for HSV-1 latency establishment (**A**) directly in DRGOs. Organoids were infected and latency was obtained through ACV addition to the culture medium. Thermal stress led to controlled reactivation (**B**), and anterograde transport of virus particles resulted in NHEK lytic infection. (**C**) Live imaging of DRGOs and NHEKs showing recombinant HSV-1 during latency (pICP0 active, green) and reactivation (pICP0 and pgC active, yellow). Scale bar, 100 μm. (**D**) Levels of ICP0 (green fluorescent signal) and gC (red fluorescent signal) expression measured as integrated density values for DRGOs and NHEKs, during latency and reactivation. Mean ± SD, * *p* < 0.05, **** *p* < 0.0001, *n* = 6 independent experiments (12 DRGOs).

**Figure 7 biomedicines-10-02068-f007:**
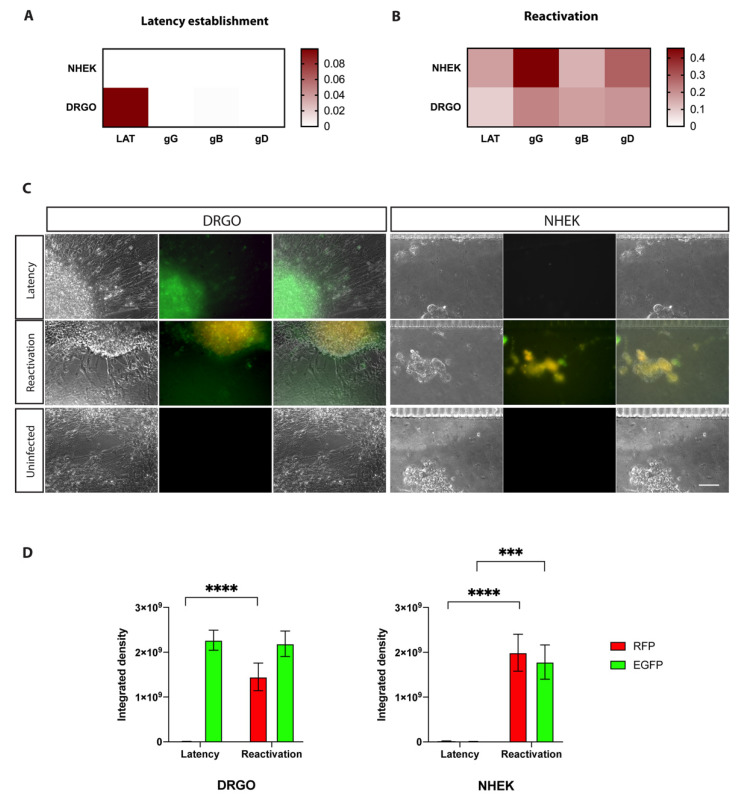
Retrograde–anterograde virus spread experiment (EXP #2). Virus gene expression profiling for HSV-1 latency establishment (**A**) indirectly in DRGOs. NHEK cells were infected, and retrograde transport of virus particles resulted in HSV-1 latency establishment in organoids. Thermal stress led to controlled reactivation (**B**), and anterograde transport of virus particles resulted in NHEK lytic infection. (**C**) Live imaging of DRGOs and NHEKs showing recombinant HSV-1 during latency (pICP0 active, green) and reactivation (pICP0 and pgC active, yellow). Scale bar, 100 μm. (**D**) Levels of ICP0 (green fluorescent signal) and gC (red fluorescent signal) expression measured as integrated density values for DRGOs and NHEKs, during latency and reactivation. Mean ± SD, *** *p* < 0.001, **** *p* < 0.0001, *n* = 6 independent experiments (12 DRGOs).

**Table 1 biomedicines-10-02068-t001:** PCR primers and size of the amplicons.

Gene	Direction	Sequence	Amplicon (bp)	Ref.
BRN3A	forward	CGTACCACACGATGAACAGC	123	
	reverse	AGGAGATGTGGTCCAGCAGA		
NAV1.7	forward	ACCTATCTCTGCTTCAAGTTGC	90	
	reverse	TGGGCTGCTTGTCTACATTAAC		
ACTB	forward	ACCCCAGCCATGTACGTT	198	
	reverse	GGTGAGGATCTTCATGAGGTAG		
ICP0-3′		TCGACCAGGGCACCCTAGT		[26]
LAT	forward	GACAGCAAAAATCCCCTGAG	192	[27]
	reverse	ACGAGGGAAAACAATAAGGG		
gG	forward	CTGTTCTCGTTCCTCACTGCCT	81	[28]
	reverse	CAAAAACGATAAGGTGTGGATGAC		
gB	forward	CCAGTCGCCAGCACAAACTCG	135	
	reverse	GCACACCACCGACCTCAAGTACAACC		
gD	forward	CCTGTCCCATCCGAACGCAGC	384	
	reverse	GCAGCAGGGTGCTCGTGTATGG		
GADPH	forward	GAATCTACTGGCGTCTTCACC	293	
	reverse	GTCATGAGCCCTTCCACGATGC		

LAT, latency-associated transcript; gG, glycoprotein G; gB, glycoprotein B; gD, glycoprotein D; GAPDH, glyceraldehyde-3-phosphate dehydrogenase.

## Data Availability

Data generated during the study are available in the NCBI GEO public repository with accession Nos. GSE133755 (https://www.ncbi.nlm.nih.gov/geo/query/acc.cgi?acc=GSE133755 (accessed on 29 April 2020)) and GSE148212 (https://www.ncbi.nlm.nih.gov/geo/query/acc.cgi?acc=GSE148212 (accessed on 29 April 2020)).

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
