# Peer review of "A Human Stem Cell-Derived Neurosensory–Epithelial Circuitry on a Chip to Model Herpes Simplex Virus Reactivation"

_biomedicines, 2022, doi:10.3390/biomedicines10092068_

Round 1
Reviewer 1 Report (Previous Reviewer 3)
The manuscript is well written and provides a reliable method to investigate the cell-cell interaction and the associated transcriptome expression. It should be published.
Reviewer 2 Report (Previous Reviewer 1)
The additions to the previous of the manuscript expanded a bit on the background issues with coculturing as requested, as well as fixing some minor text errors.
Overall the paper is in a fine condition to be accepted, there are a few minor text edits that may be of interest:
Line 31: add "they" for: from which they reactivate
Line 34: remove s from reactivations
Line 413: remove commas around next
Reviewer 3 Report (Previous Reviewer 2)
The manuscript is very well designed and written; clear and easy to understand.
This manuscript is a resubmission of an earlier submission. The following is a list of the peer review reports and author responses from that submission.
Round 1
Reviewer 1 Report
In this manuscript, the authors developed a microfluidic chip system using human stem cell derived dorsal root ganglia (DRG) organoids and applied them to herpes simplex virus (HSV) infection. They characterized the cells according to neuronal markers and transcriptional analysis for viral receptors, while the real novelty is establishing a co-culture method for neurons and epithelial cells to study viral reactivation from latency. Overall, the authors validate the system with an array of techniques and the paper should be of interest to the field as this could be a model for studying the trafficking of virions between neurons and other cells.
There are a few questions that could be elaborated in the text to better understand this model and how it overcomes previous barriers in co-culture:
What exactly in the NHEK media would be toxic, or lacking, to the neurons in a direct co-culture system?
While the authors could show there is no detectable transfer of virus between chambers under the experimental conditions, is this due to low diffusion rates through such small areas which are presumably mostly filled with the cell? Is the same true of a trackable chemical compound such as fluorescein, or if you wanted to put interferon in one well but not the other?
There are also several minor text edits that would improve readability:
Line 82: remove the comma after O.
Line 84: remove capitalization from Hepes and Fetal
Line 88: change “where” to “were”
Line 98: make the 3 in 103 superscript
Line 99: add a comma after “day”
Line 100: Change the tense, e.g., “medium was changed to KSR…”
Line 107: add a comma after DIV 10
Line 113: remove capitalization of keratinocyte
Line 128: change 64’000 to a comma
Line 143: make the 104 into a superscript exponent
Line 145: add a comma after coculture
Line 175: add a comma after briefly
Line 176: capitalize NheI
Line 204: add a comma after 2h
Line 217: add a comma after adsorption
Line 248: add a comma after chips
Lines 266-267: join 4 and °C
Line 306: add a comma after aim,
Line 314: italicize in vitro
Line 316: bold Figure S1C
Line 319: bold figure S1D-F
Line 347: italicize in vivo
Line 351: italicize in vitro
Line 450: italicize in vitro
Lines 452-460: One sentence states that CPE was visible 2 days post-reactivation, while the next sentence says the same but with 10 days. Was 2 visible for one cell type while the other took 10?
Line 460: ACV trigger, is this ACV removal? Heat stress is the trigger
Line 463: add a comma after removal
Line 643: remove capitalization from Chip
Line 649: change differently to different
Line 670: remove comma after assays
Author Response
In this manuscript, the authors developed a microfluidic chip system using human stem cell derived dorsal root ganglia (DRG) organoids and applied them to herpes simplex virus (HSV) infection. They characterized the cells according to neuronal markers and transcriptional analysis for viral receptors, while the real novelty is establishing a co-culture method for neurons and epithelial cells to study viral reactivation from latency. Overall, the authors validate the system with an array of techniques and the paper should be of interest to the field as this could be a model for studying the trafficking of virions between neurons and other cells.
There are a few questions that could be elaborated in the text to better understand this model and how it overcomes previous barriers in co-culture:
What exactly in the NHEK media would be toxic, or lacking, to the neurons in a direct co-culture system?
The exact NHEK medium composition is unknown, as it available as a “ready to use” formulation by the Supplier. However, CaCl2 0.06 mM is reported as a supplement, while in DMEM, used for Hacat cells co-culture, the final concentration is 1.8 mM. This lower than physiological CaCl2 is used to maintain keratinocytes in an undifferentiated state which allows them to proliferate. This might have impaired DRGO survival, as it has already described by Belote and Simon (https://doi.org/10.1083/jcb.201902014). In detail, they described how mature co-cultures were obtained only increasing the CaCl2 concentration of the growth media from 0.06 mM CaCl2 to 1.06 mM CaCl2. The main text was implemented thanks to the reviewer’s suggestion, and we added the proper reference (Line 652, Ref 44).
While the authors could show there is no detectable transfer of virus between chambers under the experimental conditions, is this due to low diffusion rates through such small areas which are presumably mostly filled with the cell? Is the same true of a trackable chemical compound such as fluorescein, or if you wanted to put interferon in one well but not the other?
Taylor et al. demonstrated that the two chambers of a chip developed with microchannels of the dimensions we chose and used for axonal growth were independent due to the high fluidic resistance. In detail, the authors did not detect Texas red dextran (3,000 Da; 20 μM) for over 20 hours on the other side of the chip (https://doi.org/10.1038/nmeth777). Thus, as the molecular weight values of IFNs range from 32,000 to 73,000 Da, these proteins would be subjected to the same fluidic resistance and prevented from spreading between the chip chambers. The main text and references were updated thanks to the reviewer’s comment (Lines 444-445, Ref. 36).
There are also several minor text edits that would improve readability:
Line 82: remove the comma after O.
We corrected the manuscript accordingly to the reviewer’s comments.
Line 84: remove capitalization from Hepes and Fetal
We corrected the manuscript accordingly to the reviewer’s comments.
Line 88: change “where” to “were”
We corrected the manuscript accordingly to the reviewer’s comments.
Line 98: make the 3 in 103 superscript
We corrected the manuscript accordingly to the reviewer’s comments.
Line 99: add a comma after “day”
We corrected the manuscript accordingly to the reviewer’s comments.
Line 100: Change the tense, e.g., “medium was changed to KSR…”
We corrected the manuscript accordingly to the reviewer’s comments.
Line 107: add a comma after DIV 10
We corrected the manuscript accordingly to the reviewer’s comments.
Line 113: remove capitalization of keratinocyte
We corrected the manuscript accordingly to the reviewer’s comments.
Line 128: change 64’000 to a comma
We corrected the manuscript accordingly to the reviewer’s comments.
Line 143: make the 104 into a superscript exponent
We corrected the manuscript accordingly to the reviewer’s comments.
Line 145: add a comma after coculture
We corrected the manuscript accordingly to the reviewer’s comments.
Line 175: add a comma after briefly
We corrected the manuscript accordingly to the reviewer’s comments.
Line 176: capitalize NheI
We corrected the manuscript accordingly to the reviewer’s comments.
Line 204: add a comma after 2h
We corrected the manuscript accordingly to the reviewer’s comments.
Line 217: add a comma after adsorption
We corrected the manuscript accordingly to the reviewer’s comments.
Line 248: add a comma after chips
We corrected the manuscript accordingly to the reviewer’s comments.
Lines 266-267: join 4 and °C
We corrected the manuscript accordingly to the reviewer’s comments.
Line 306: add a comma after aim,
We corrected the manuscript accordingly to the reviewer’s comments.
Line 314: italicize in vitro
We corrected the manuscript accordingly to the reviewer’s comments.
Line 316: bold Figure S1C
We corrected the manuscript accordingly to the reviewer’s comments.
Line 319: bold figure S1D-F
We corrected the manuscript accordingly to the reviewer’s comments.
Line 347: italicize in vivo
We corrected the manuscript accordingly to the reviewer’s comments.
Line 351: italicize in vitro
We corrected the manuscript accordingly to the reviewer’s comments.
Line 450: italicize in vitro
We corrected the manuscript accordingly to the reviewer’s comments.
Lines 452-460: One sentence states that CPE was visible 2 days post-reactivation, while the next sentence says the same but with 10 days. Was 2 visible for one cell type while the other took 10?
The reviewer is right. CPE was visible 2 days post-reactivation in SH-SY5Y and 10 days post-reactivation in DRGOs. We corrected the manuscript accordingly to the reviewer’s comments (Lines 457, 460)
Line 460: ACV trigger, is this ACV removal? Heat stress is the trigger
It is ACV treatment. We corrected the manuscript accordingly to the reviewer’s comments.
Line 463: add a comma after removal
We corrected the manuscript accordingly to the reviewer’s comments.
Line 643: remove capitalization from Chip
We corrected the manuscript accordingly to the reviewer’s comments.
Line 649: change differently to different
We corrected the manuscript accordingly to the reviewer’s comments.
Line 670: remove comma after assays
We corrected the manuscript accordingly to the reviewer’s comments.
Reviewer 2 Report
The manuscript is very well written; clear, precise, and easy to understand with lots of explanatory data.
Please just re-check and correct the usage of abbreviations in some places (ex. dorsal root ganglia organoids; DRGOs).
Author Response
The manuscript is very well written; clear, precise, and easy to understand with lots of explanatory data.
Please just re-check and correct the usage of abbreviations in some places (ex. dorsal root ganglia organoids; DRGOs).
We thank the reviewer for the kind comments. We corrected the manuscript accordingly to the reviewer’s suggestions.
Reviewer 3 Report
This is a very interesting and innovative method to design two separate culture condition connected by the microfluid system to investigate cell-cell interaction.
This study revealed a topographic relation mimicking the sensory-epithelial connection on-a-chip assay in iPS dorsal root ganglion cells and keratinocyte. The results showed that the bulk and single cell transcriptomics expressed in each sensory neuronal cell type. This neuronal- epithelial circuitry provided a detailed analysis of the HSV infectivity modeling either its dynamics and cell type specificity. The reconstitution of an organized connectivity between human sensory neurons and keratinocytes into microfluidic chips provides a powerful in vitro platform to model viral latency and reactivation of human viral pathogens.
The manuscript is well written and provides a reliable method to investigate the cell-cell interaction and the associated transcriptome expression. It should be published.
Author Response
This is a very interesting and innovative method to design two separate culture condition connected by the microfluid system to investigate cell-cell interaction.
This study revealed a topographic relation mimicking the sensory-epithelial connection on-a-chip assay in iPS dorsal root ganglion cells and keratinocyte. The results showed that the bulk and single cell transcriptomics expressed in each sensory neuronal cell type. This neuronal- epithelial circuitry provided a detailed analysis of the HSV infectivity modeling either its dynamics and cell type specificity. The reconstitution of an organized connectivity between human sensory neurons and keratinocytes into microfluidic chips provides a powerful in vitro platform to model viral latency and reactivation of human viral pathogens.
The manuscript is well written and provides a reliable method to investigate the cell-cell interaction and the associated transcriptome expression. It should be published.
We thank the reviewer for the kind comments.